# Proteomic Profiling of Endothelial Cell Secretomes After Exposure to Calciprotein Particles Reveals Downregulation of Basement Membrane Assembly and Increased Release of Soluble CD59

**DOI:** 10.3390/ijms252111382

**Published:** 2024-10-23

**Authors:** Alexander Stepanov, Daria Shishkova, Victoria Markova, Yulia Markova, Alexey Frolov, Anastasia Lazebnaya, Karina Oshchepkova, Daria Perepletchikova, Daria Smirnova, Liubov Basovich, Egor Repkin, Anton Kutikhin

**Affiliations:** 1Department of Experimental Medicine, Research Institute for Complex Issues of Cardiovascular Diseases, 6 Barbarash Boulevard, 650002 Kemerovo, Russia; stepav@kemcardio.ru (A.S.); shidk@kemcardio.ru (D.S.); markve@kemcardio.ru (V.M.); markyo@kemcardio.ru (Y.M.); frolav@kemcardio.ru (A.F.); lazeai@kemcardio.ru (A.L.); oschki@kemcardio.ru (K.O.); 2Laboratory of Regenerative Biomedicine, Institute of Cytology of the RAS, 4 Tikhoretskiy Prospekt, 194064 St. Petersburg, Russia; perepletchikova@incras.ru (D.P.); smirnova@incras.ru (D.S.); basovich@incras.ru (L.B.); 3Resource Centre for Molecular and Cell Technologies, St. Petersburg State University, Universitetskaya Embankment, 7/9, 199034 St. Petersburg, Russia; st049553@student.spbu.ru

**Keywords:** endothelial cells, endothelial dysfunction, calciprotein particles, calcium stress, endothelial secretome, extracellular matrix, basement membrane, CD59, proteomic profiling, bioinformatic analysis

## Abstract

Calciprotein particles (CPPs) are essential circulating scavengers of excessive Ca^2+^ and PO_4_^3−^ ions, representing a vehicle that removes them from the human body and precludes extraskeletal calcification. Having been internalised by endothelial cells (ECs), CPPs induce their dysfunction, which is accompanied by a remarkable molecular reconfiguration, although little is known about this process’s extracellular signatures. Here, we applied ultra-high performance liquid chromatography-tandem mass spectrometry to perform a secretome-wide profiling of the cell culture supernatant from primary human coronary artery ECs (HCAECs) and internal thoracic artery ECs (HITAECs) treated with primary CPPs (CPP-P), secondary CPPs (CPP-S), magnesiprotein particles (MPPs), or Ca^2+^/Mg^2+^-free Dulbecco’s phosphate-buffered saline (DPBS) for 24 h. Incubation with CPP-P/CPP-S significantly altered the profiles of secreted proteins, delineating physiological and pathological endothelial secretomes. Neither pathway enrichment analysis nor the interrogation of protein–protein interactions detected extracellular matrix- and basement membrane-related molecular terms in the protein datasets from CPP-P/CPP-S-treated ECs. Both proteomic profiling and enzyme-linked immunosorbent assay identified an increased level of protectin (CD59) and reduced levels of osteonectin (SPARC), perlecan (HSPG2), and fibronectin (FN1) in the cell culture supernatant upon CPP-P/CPP-S treatment. Elevated soluble CD59 and decreased release of basement membrane components might be considered as potential signs of dysfunctional endothelium.

## 1. Introduction

Quiescent endothelial cells (ECs) constitute an anti-thrombotic, anti-inflammatory, and vasoprotective layer governing vascular homeostasis [1,2]. Endothelial functioning becomes significantly impaired in metabolic disorders such as dyslipidemia (which is particularly common in overweight and obese patients), hyperglycemia (which defines the development of diabetes mellitus), uremia and hyperphosphatemia (which are inevitable consequences of chronic kidney disease and particularly end-stage renal disease), or hypercalcemia (which accompanies hyperparathyroidism) [3,4,5,6]. In these pathological conditions and disease states, ECs undergo a pro-inflammatory activation notable for augmented release of the respective cytokines, overexpression of cell adhesion molecules responsible for monocyte attachment, and increased endothelial permeability [3,4,5,6]. Taken together, these molecular consequences facilitate lipid retention and acidification of the intimal microenvironment, thereby acting as a prerequisite for the development of atherosclerosis [7,8]. Further, excessive release of endothelial-derived cytokines (e.g., interleukin (IL)-6, interleukin-8, and monocyte chemoattractant protein 1/chemokine (C-C motif) ligand 2 (MCP-1/CCL2)) contributes to chronic low-grade inflammation, which represents a molecular basis for inflammageing [9,10]. Moreover, elevated levels of pro-inflammatory cytokines emerge as an independent trigger of endothelial dysfunction, thus creating a vicious cycle in the elderly [11,12], patients with severe COVID-19 [13,14] or sepsis [15,16,17,18], and those with multiple organ dysfunction syndrome [19].

Calciprotein particles (CPPs) are assembled in the human blood from calcium, phosphate, and acidic serum proteins (primarily fetuin-A and albumin) and operate as inherent scavengers, removing excessive minerals from the circulation [20,21,22,23]. Initially having a spherical form and an amorphous structure, primary CPPs (CPP-P) undergo gradual conversion to secondary CPPs (CPP-S), which acquire a needle or spindle shape in conjunction with high crystallinity [20,21,22,23]. Although the generation of CPPs is a vital process that is essential to maintaining mineral homeostasis, their engulfment by macrovascular ECs [24,25,26,27,28,29,30,31], liver sinusoidal ECs [32,33], monocytes [31], liver or spleen macrophages [32,33,34,35], and hepatocytes [35] promotes their pathological activation and contributes to low-grade inflammation in the elderly [36], patients with disturbed mineral homeostasis [27], and those with a systemic disease and ≥1 comorbid condition [37]. Previous studies showed that the internalisation of CPP-P or CPP-S by human coronary artery endothelial cells (HCAEC) and human internal thoracic artery endothelial cells (HITAEC) induced the release of IL-6, IL-8, and MCP-1/CCL2—as well as the overexpression of vascular cell adhesion molecule 1 (VCAM1) and intercellular cell adhesion molecule 1 (ICAM1)—as a result of CPP dissolution in lysosomes, uncontrolled Ca^2+^ burst in the cytosol, NLRP3 inflammasome activation [32,33,38,39], and upregulation of the corresponding genes [24,25,26,27,28,29,30,31]. Further, dot blot profiling of EC culture supernatant revealed an increased release of serpin E1 (plasminogen activator inhibitor 1, PAI-1), urokinase plasminogen activator surface receptor (uPAR), chemokine (C-X-C motif) ligand 1/growth regulated protein alpha (CXCL1/GROα, and macrophage inflammatory protein-3 alpha/chemokine (C-C motif) ligand 20 (MIP-3α/CCL20)) upon CPP treatment in both HCAECs and HITAECs [40].

However, the EC secretome contains from 1550 to 2650 proteins [41,42], most of which have not been measured after CPP exposure. Besides pro-inflammatory cytokines, ECs produce a myriad of molecules with pro- and anti-thrombotic, pro- and anti-fibrotic, and vasoactive effects [40,41,42,43,44]. An intricate interplay between EC-secreted proteins modulates their paracrine and endocrine effects, which has a significant impact on the vascular environment and systemic homeostasis [40,43]. For instance, beneficial interactions between HCAECs and HITAECs justify multiple arterial graftings and total arterial revascularisations, providing an argument for the complete avoidance of using the saphenous vein as a bypass [40,43]. A proteome-wide analysis of normal and pathological EC secretomes is a compulsory prerequisite for the identification of reliable endothelial dysfunction markers, which remains an unmet need in pathophysiology and clinical practice.

The integrity of the basement membrane (BM) has been suggested as a requirement to maintain endothelial homeostasis, including the physiological profile of secreted molecules [45]. The subendothelial extracellular matrix (ECM) modulates the activation and inhibition of biochemical pathways in a composition-specific (i.e., dependent on the ratio between different components) and flow-specific (i.e., shear stress-dependent) manner [46]. The BM composition determines cell survival and significantly impacts cell metabolism under various stress conditions, e.g., in a pro-inflammatory microenvironment [47]. Current vascular tissue engineering employs a number of in vitro strategies for recapitulating the BM or retaining its function after the decellularisation in order to meet regenerative medicine research needs, although an ideal approach is yet to be developed [48]. The success of these endeavours might significantly improve endothelialisation of vascular grafts and implantation results, reducing the risk of thrombosis, neointimal hyperplasia, and neoatherosclerosis [48]. To summarise, the qualitative and quantitative composition of the BM is one of the primary factors affecting endothelial physiology.

A factor that has a major impact on the protection of host cells from the membrane attack complex formation, complement-mediated osmotic lysis, and respective autoimmune damage (e.g., demyelinating neuropathy) is the surface glycoprotein CD59, also termed protectin [49,50]. CD59 is broadly expressed in the human body, and its deficiency in red blood cells leads to their complement-dependent lysis, followed by haemolytic anemia [51] and paroxysmal nocturnal hemoglobinuria [52]. The protective mechanism of CD59 action implies competitive binding to neoepitopes on C5b-8/9, thus limiting its binding to C9 and preventing the C5b-8/9-catalysed insertion of C9 into the lipid bilayer [53,54]. In other studies, the specific binding of CD59 to C8a and C9b subunits has been both suggested [55] and detected [56]. Notably, functional CD59 can be conveyed from red blood cells to ECs through intermembrane transfer, reinforcing their autoprotective capabilities [57].

The glycation of CD59 (presumably its Lys41/His44 motif [58]) leads to its inactivation and results in its inability to perform protection against complement-dependent lysis, explaining the deposition of the membrane attack complex on blood vessels [59] and red blood cells [60] and subsequent vascular injury in target organs (such as the kidneys and nerves) [59] and haemolytic anemia [60] in patients with diabetes. Mutated CD59 has a higher susceptibility to glycation than the wild-type receptor, especially under hyperglycemic conditions [61], and inherent CD59 deficiency might lead to an early-onset haemolytic phenotype causing angiopathy and polyneuropathy [62,63]. Strikingly, CD59-deficient hyperlipidemic mice were characterised by endothelial injury, accelerated atherosclerosis, higher plaque burden, and co-localisation of the membrane attack complex with atherosclerotic lesions [64,65,66,67,68,69], whilst CD59 overexpression or complement inhibition attenuated plaque development [65]. In hyperlipidemic mice, the combination of diabetes and CD59 deficiency promoted atherosclerosis in comparison with diabetes alone [68]. The administration of C-phycocyanin, which increases CD59 expression [70], pharmacological inhibition of complement [71], or liposome-mediated gene therapy with CD59 [72] inhibited endothelial apoptosis, curbed vascular smooth muscle cell proliferation, reduced blood lipid levels, and mitigated atherosclerotic burden in ApoE-knockout mice [70,71,72].

Plasma glycated CD59, detectable via a sensitive and specific enzyme-linked immunosorbent assay [73], has been suggested as a promising diagnostic [73,74,75,76,77] and predictive [78] biomarker of type 2 diabetes [73], pregnancy-induced glucose intolerance [74], gestational diabetes [75,76,77,78], and postpartum glucose intolerance [79,80]. Moreover, plasma glycated CD59 predicted adverse pregnancy outcomes, such as large-for-gestational-age newborns [74,77,81], pregnancy-induced hypertension [81], and neonatal hypoglycemia [81,82]. Plasma glycated CD59 was directly correlated with increased blood glucose after the oral glucose tolerance test and glycated haemoglobin (HbA1c) [83]. Likewise, the lowering of plasma glycated CD59 was intimately associated with a reduction in blood glucose [83], whereas intracellular (i.e., non-secreted) CD59 is required for normal insulin secretion by pancreatic β cells and is downregulated in patients with diabetes [84]. Hence, shedding and/or glycation of CD59 from endothelial cells might represent a pathological event indicative of endothelial dysfunction and vulnerability to its triggers, such as calcium stress, uremia, or elevated amounts of pro-inflammatory cytokines in the milieu.

Here, we applied ultra-high performance liquid chromatography-tandem mass spectrometry (UHPLC-MS/MS, TimsToF Pro mass spectrometer, Bruker Daltonics, Billerica, MA, USA) to perform an unbiased, high-throughput, label-free proteomic analysis of the secretome collected from CPP-P- and CPP-S-treated HCAECs and HITAECs. Exposure to CPP-P or CPP-S diminished release of the ECM—in particular, of the BM constituents into the milieu—and promoted the liberation of soluble CD59 (a complement-protecting cell surface glycoprotein receptor), whilst molecular signatures of vasospastic, pro-thrombotic, and pro-fibrotic activation have not been found. Pathway enrichment and protein–protein interaction analysis identified four groups of downregulated molecules, including those responsible for ECM synthesis, protein folding, transcription and translation, and the cytoskeleton. This pattern recapitulated the molecular reconfiguration of aging endothelium and suggested the role of calcium stress-induced endothelial dysfunction in promoting endothelial senescence. Incubation of ECs with either CPP-P or CPP-S induced similar molecular consequences, whereas magnesiprotein particles (MPPs) as expected did not induce any pathological response. The relative distance between the secretomes of CPP-P/CPP-S- and DPBS/MPP-treated ECs was considerably higher than between those of intact EC lines, suggesting a major impact of endothelial dysfunction on endothelial heterogeneity. Taken together, these findings advance our knowledge of endothelial dysfunction triggered by calcium stress and suggest increased release of soluble CD59 and impaired export of BM components as putative features of dysfunctional endothelium.

## 2. Results

To analyse the differences between the physiological and pathological secretomes of HCAECs and HITAECs under calcium stress, we replaced the complete cell culture medium with a serum-free medium upon reaching cell confluence. Then, we incubated HCAECs and HITAECs with CPP-P or CPP-S (25 µg/µL calcium), calcium-free MPP, or Ca^2+^ and Mg^2+^-free Dulbecco’s phosphate-buffered saline (DPBS, control solution) for 24 h, with consecutive centrifugation of cell culture supernatant at 220× *g* to sediment detached cells and at 2000× *g* to sediment the cell debris. Before conducting a proteomic analysis, we performed an extensive screening of pro-inflammatory cytokines in the cell culture supernatant to ensure that CPP-P and CPP-S indeed induced endothelial activation. Dot blotting analysis revealed excessive release of multiple pro-inflammatory cytokines into the cell culture medium by HCAECs and HITAECs upon CPP-P or CPP-S treatment (Figure 1). Among the most prominent upregulated cytokines were interleukin-6 (IL-6), chemokine (C-C motif) ligand 20 (CCL20)/macrophage inflammatory protein-3 alpha (MIP-3α), CCL5/regulated on activation, normal T cell expressed and secreted (RANTES), and soluble CD105/endoglin, a detached form of the EC receptor indicating cell death, although soluble CD31/PECAM1 was not overrepresented after the incubation with CPPs (Figure 1). This verified the development of pro-inflammatory endothelial dysfunction upon treatment with CPP-P or CPP-S, characterised by the increased production of respective cytokines and induction of apoptotic cascades in a certain proportion of ECs.

Upon protein precipitation, denaturation, and trypsinisation, peptides were desalted and evaluated via UHPLC-MS/MS, followed by a bioinformatic analysis. In total, we identified 1246 proteins that were secreted in ≥15/22 (≥70%) of samples. Principal component analysis showed a clusterisation of physiological (i.e., DPBS- and MPP-treated) and pathological (i.e., CPP-P- and CPP-S-treated) protein profiles of secreted molecules (Figure 2A). The magnitude of difference between the control and experimental groups was largely similar between HCAECs and HITAECs (Figure 2A). The number of unique differentially expressed proteins (i.e., those with logarithmic fold change ≥ 1 and false discovery rate-corrected *p* value ≤ 0.05) was significant across each of the experimental (CPP-P or CPP-S) to control (DPBS or MPP) group comparisons, indicating a particle-specific molecular response to CPP-P and CPP-S in both of the EC lines (Figure 2B).

Analysis of CPP-P vs. DPBS, CPP-P vs. MPP, CPP-S vs. DPBS, and CPP-S vs. MPP comparisons for each of the EC lines detected higher numbers of underexpressed proteins in comparison with overexpressed proteins upon CPP-P or CPP-S treatment (Table 1 and Appendix A). Inspection of MPP vs. DPBS comparisons found 54 and 20 differentially expressed proteins in HCAECs and HITAECs, respectively (Table 1 and Appendix A). Inversely, examination of CPP-S vs. CPP-P comparisons demonstrated 8 and 89 differentially expressed proteins in HCAECs and HITAECs, respectively (Table 1 and Appendix A). However, pathway enrichment analysis of upregulated and downregulated proteins in both DPBS vs. MPP and CPP-S vs. CPP-P comparisons showed indecisive results. Hence, we pooled the DPBS/MPP and CPP-P/CPP-S groups to generalise the conclusions on physiological and pathological EC secretomes.

First, we itemised the 50 most abundant proteins in each of the pooled samples, showing 36 and 24 specific proteins between DPBS/MPP and CPP-P/CPP-S groups in HCAECs and HITAECs, respectively (Figure 3A). In total, 5 proteins (CD59, RPS27A, WDR1, TALDO1, and GPI) and 7 proteins (SPARC, HSPG2, CCN2, TPM4, IGFBP7, CALM1, and HSP90AA1) were represented exclusively amongst the top 50 in the pathological and physiological secretome, respectively (Figure 3A). Enrichment with exosomal (≈75–85%) and extracellular space (≈50–55%) proteins confirmed successful purification during the sample preparation, whilst enrichment with plasma membrane proteins (≈40–50%) suggested a considerable amount of protein shedding into the milieu (Figure 3B). Among the most abundant proteins in the physiological secretome were those responsible for the following: (1) hemostasis (≈38–40%), as well as platelet activation, signalling and aggregation (≈31–35%); (2) innate immune response (≈35–37%), cytokine signalling (≈20–22%) and interleukin signalling (≈17%); (3) focal adhesion (≈26%) and cytoskeleton (≈18–22%); (4) collagen-containing ECM (≈24–30%), ECM organisation (≈15–17%), and ECM structural constituents (≈11%); (5) angiogenesis (≈11–15%) and VEGF signalling (≈11%) (Figure 3B). In contrast, the most abundant proteins in the pathological secretome did not belong to ECM- and angiogenesis-related categories (excluding collagen-containing ECM in the HITAEC secretome, ≈17% proteins, Figure 3B). Proteins accountable for hemostasis and platelet activation, signalling, and aggregation were also less represented in the pathological secretome (≈29–35% and ≈21–27%, respectively, Figure 3B). Hence, we suggest that the pathological endothelial secretome under calcium stress is characterised by a reduced synthesis of ECM components.

To test this hypothesis, we carried out a pathway enrichment analysis of secreted proteins that were underexpressed in the cell culture supernatant upon the treatment of HCAECs and HITAECs with CPP-P or CPP-S. Several molecular terms related to the ECM were overrepresented in these datasets, including “extracellular matrix”, “collagen-containing extracellular matrix”, “extracellular matrix organization”, “extracellular matrix structural constituent”, “degradation of the extracellular matrix”, “collagen formation”, and “basement membrane” (Figure 4A). As BM and subendothelial ECM integrity are essential prerequisites for proper endothelial functioning, we further focused on comparing the abundance of respective proteins in DPBS/MPP and CPP-P/CPP-S clusters. Osteonectin (secreted protein acidic and rich in cysteine, SPARC), perlecan (heparan sulfate proteoglycan 2, HSPG2), fibronectin (FN1), tissue inhibitor of metalloproteinase 1 (TIMP1), a number of laminin subunits (LAMB1, LAMA4, LAMC1), lysyl oxidase-like 2 (LOXL2), nidogen 1 (NID1), fibrillin-1 (FBN1), agrin (AGRN), collagen 4 and 6 subunits (COL4A2, COL6A1), and peroxidasin (PXDN) were among the significantly downregulated BM components (Figure 4B). Notably, biglycan (BGN), osteonectin (SPARC), perlecan (heparan sulfate proteoglycan 2, HSPG2), and thrombospondin-1 (THBS1) were presented in the 10 most downregulated proteins upon CPP-P or CPP-S treatment in most of the comparisons (6 or 7 out of 8, Table 2). Similar analysis among the upregulated proteins did not reveal any ECM-related molecular terms. Strikingly, soluble protectin (CD59) was the only protein included in the 10 most upregulated proteins in all comparisons after the incubation of ECs with CPP-P or CPP-S, whereas the most abundant endothelial pro-inflammatory cytokine, macrophage migration inhibitory factor (MIF), and extracellular vesicle markers caveolin-1 (CAV1) and CD63 were inconsistently presented in this list (Table 2).

We next analysed protein–protein interactions within the proteins underrepresented in the cell culture supernatant from HCAECs and HITAECs upon CPP-P or CPP-S treatment. Screening using the Search Tool for the Retrieval of Interacting Genes/Proteins (STRING) identified four clusters of interacting proteins, which included those responsible for (1) ECM production, (2) protein folding, (3) translation and transcription, and (4) the cytoskeleton (Figure 5 and Figure 6). An additional minor cluster of interactions within the 14-3-3 signalling pathway was also revealed (Figure 5 and Figure 6). A similar analysis of protein–protein interactions among the upregulated proteins did not detect any consistent groups of interacting proteins, in accordance with previous findings (Appendix A). Examination of protein–protein interactions in relation to MPP vs. DPBS and CPP-S vs. CPP-P comparisons also did not provide any definitive results (Appendix A).

We further compared the extent of inducible (i.e., triggered by CPP-P or CPP-S) and constitutive (i.e., baseline) endothelial heterogeneity in relation to the profiles of secreted molecules. Principal component analysis of all the secretomes clearly showed three clusters: (1) DPBS/MPP-treated ECs; (2) CPP-P/CPP-S-treated HCAEC; (3) CPP-P/CPP-S-treated HITAECs (Figure 7A). Pooling of the DPBS and MPP groups together identified a certain distance between HCAEC and HITAEC secretomes (Figure 7B). The datasets of differentially expressed proteins across the intergroup comparisons (MPP vs. DPBS, CPP-S vs. CPP-P, CPP-P vs. DPBS, CPP-P vs. MPP, CPP-S vs. DPBS, and CPP-S vs. MPP) showed a minor overlap between EC lines, which indicates cell-specific molecular reconfiguration patterns in response to CPP-P or CPP-S (Figure 7C). Yet, although we documented 65 differentially expressed proteins between the secretomes of intact (DPBS/MPP-treated) HCAECs and HITAECs (43 proteins overexpressed in HCAECs and 22 proteins upregulated in HITAECs, Figure 7D and Appendix A), they demonstrated insignificant differences in protein abundance between EC types (Figure 7E). Analysis of protein–protein interactions also did not specify any molecular terms enriched in these protein datasets (Appendix A).

Finally, we performed an enzyme-linked immunosorbent assay (ELISA) against protectin (CD59, which was consistently upregulated upon CPP-P or CPP-S treatment) and the three most abundant downregulated BM components—osteonectin (SPARC), perlecan (HSPG2), and fibronectin (FN1—to verify the results of the proteomic profiling. The levels of protectin (CD59) were significantly higher, and the concentrations of osteonectin (SPARC), perlecan (HSPG2), and fibronectin (FN1) were significantly lower in the cell culture supernatant from CPP-P- and CPP-S-treated HCAECs and HITAECs, in comparison with those incubated with DPBS or MPP (Figure 8). Notably, the relative levels of soluble endoglin/CD105 and PECAM1/CD31 in the experimental groups (CPP-P/CPP-S vs. DPBS/MPP) corresponded between the dot blot and proteomics analysis, and also between each other (the level of PECAM1/CD31 was higher than the level of endoglin/CD105) (Figure 1 and Appendix A).

## 3. Discussion

Calciprotein particles (CPPs), assembled as a result of molecular interactions between fetuin-A and nascent calcium phosphate clusters, are indispensable scavengers of excessive Ca^2+^ and PO_4_^3−^ ions, thus representing an elegant mechanism of regulating mineral homeostasis [20,21,22,23]. The generation of CPPs represents an evolutionary strategy to prevent blood supersaturation with Ca^2+^ and PO_4_^3−^ ions (e.g., as a result of bone resorption) and to thwart extraskeletal calcification, a pathological condition which is frequent in patients with chronic kidney disease [20,21,22,23]. It is believed that the formation of CPPs accompanied evolution from the appearance of bony fish; the existence of CPPs in more ancient animals remains uncertain. Shortly after their emergence, amorphous, spherical, and submicrometer-sized (30–100 nm) primary CPPs (CPP-P) evolve into crystalline-, spindle- or needle-shaped, micrometer-sized (100–300 nm) secondary CPPs (CPP-S) [20,21,22,23]. Upon executing their function of aggregating Ca^2+^ and PO_4_^3−^ ions, CPPs are removed from the circulation by macrovascular [24,25,26,27,28,29,30,31] or liver sinusoidal ECs [32,33], monocytes [31], liver or spleen macrophages [32,33,34,35], and hepatocytes [35]. The internalisation and digestion of CPPs by ECs induce a chain of detrimental events, including an increase in cytosolic Ca^2+^, mitochondrial and endoplasmic reticulum stress, nuclear factor (NF)-κB-mediated transcriptional response, and cytokine release [24,26,27,30,31,32,33,38,39], ultimately contributing to the development of pro-inflammatory endothelial activation, chronic low-grade inflammation, and inflammageing [27,29,31]. The pathological permeability of dysfunctional endothelium promotes lipid retention and leukocyte extravasation, together contributing to the development of vascular inflammation, proteolytic environment, degradation of the BM and internal elastic lamina, and contractile-to-synthetic switch of vascular smooth muscle cells [4,5,6,7,8,12]. Such an ensemble of molecular events ultimately induces intimal hyperplasia, vascular remodeling, and atherosclerotic progression [4,5,6,7,8,12]. Hence, CPPs act as a double-edged sword that rescue the human body from an acute life-threatening disease (i.e., extraskeletal calcification) at the cost of promoting long-lasting pathological conditions (i.e., endothelial dysfunction and atherosclerosis).

Among the several forms of calcium transfer (free Ca^2+^ ions, colloidal calciprotein monomers, and particulate CPPs), CPPs are the most efficient in delivering calcium stress to ECs because of their deposition and dissolution in lysosomes, followed by lysosomal membrane permeabilisation and an excessive Ca^2+^ migration into the cytosol [27,32,38,39]. The molecular pattern of CPP-induced calcium stress within the ECs is well defined and includes elevated expression of cell adhesion molecules (VCAM1, ICAM1, and E-selectin) and endothelial-to-mesenchymal transition transcription factors (SNAI1, SNAI2, TWIST1, and ZEB1) in combination with endothelial nitric oxide synthase (eNOS) uncoupling [25,26,27,30,31]. However, our knowledge of extracellular signatures of such endothelial responses remains limited to an augmented release of pro-inflammatory cytokines (IL-6, IL-8, MCP-1/CCL2, MIF, CXCL1, and MIP-3α/CCL20) and pro- or anti-thrombotic molecules (serpin E1/PAI-1 and uPAR) [24,27,31]. As EC-secreted bioactive factors (termed angiokines) control vascular tone, maintain haemostasis, and regulate the instructive signalling governing local homeostasis within most organs [1,2], the decryption and interpretation of the endothelial secretome in physiological and pathological conditions are of utmost importance. The discovery of circulating biomarkers specific for endothelial dysfunction, which can be measured by a routine enzyme-linked immunosorbent assay, might inform us on specific disease states and prompt the respective pharmacological interventions.

Here, we aimed at deciphering the secretome of ECs treated with either CPP-P or CPP-S, employing UHPLC-MS/MS for the analysis of serum-free cell culture supernatant. For the objective comparison of heterogeneous arterial ECs, we utilised human coronary artery endothelial cells (HCAECs) and human internal thoracic artery endothelial cells (HITAECs), which comprise the innermost lining in atheroprone and atheroresistant blood vessels (the coronary artery and internal thoracic artery, respectively). We found that treatment with CPP-P or CPP-S curtailed the production of ECM proteins, in particular BM components. These findings were in accordance with other studies that documented progressive thinning of the BM in vascular disease [85,86] and detected reduced release of its constituents after the exposure of ECs to environmental triggers of endothelial dysfunction, such as sidestream tobacco smoke [87], lead [88], high glucose [89,90], or IL-6 [91]. All BM components, which were underexpressed in the extracellular milieu upon treatment with CPP-P or CPP-S (i.e., osteonectin, perlecan, fibronectin, laminin subunit alpha 4, laminin subunit beta 1, laminin subunit gamma 1, collagen type IV alpha 2 chain, collagen type VI alpha 1 chain, nidogen 1, fibrillin-1, agrin, tissue inhibitor of metalloproteinase 1, and lysyl oxidase-like 2), are considered as essential for BM assembly [92,93,94,95,96,97,98,99,100,101]. Among the primary BM proteins are type IV collagen, laminin, nidogen, and perlecan [92,93,94,95,96,97,98,99,100,101]. Collagen type IV and laminin form two self-assembling supramolecular networks that are linked together by nidogen and perlecan bridges [92,93,94,95,96,97,98,99,100,101]. Each of these components also impacts endothelial physiology in vitro [92,93,94,95,96,97,98,99,100,101]. For instance, perlecan deficiency promoted endothelial dysfunction by lowering the expression of endothelial nitric oxide synthase and reducing endothelial-dependent vasorelaxation [102], whilst perlecan itself enhanced angiogenesis and wound healing [103]. Defective BMs and disordered orientation of ECs are common electron microscopy signs of dysfunctional endothelium [104] and are associated with impaired mechanotransduction [105,106,107] and vascular ageing [108]. Notably, treatment with calciprotein particles compromises mechanosensing through the downregulation of atheroprotective transcription factors KLF2 and KLF4, along with the derepression of the activity of YAP1, an atherogenic downstream effector of the Hippo signaling pathway [26].

In a recent proteomic study, we found that treatment of HCAECs and HITAECs with CPP-P or CPP-S dwindled transcriptional, post-transcriptional, and translational activity, impeded protein folding, inhibited amino acid metabolism, and reduced energy generation [30]. Here, the proteins which were underrepresented in the secretome upon CPP-P or CPP-S treatment formed distinct molecular interaction clusters in transcription/translation and protein folding domains. Hence, incubation with CPP-P or CPP-S hampered endothelial metabolism, which was in concert with the lessened synthesis of the ECM and BM components. Such a molecular pattern suggests weakened endothelial resilience (i.e., reduced molecular and phenotypical adaptation to the altered microenvironment) [109,110] as a typical consequence of calcium stress. This corresponds to the pathophysiological concepts that propose a synergistic action among endothelial dysfunction triggers [4,5,12], thus emphasizing the need to apply the respective pharmacological approaches to correct the specific risk factors.

As in our previous study [31], the extent of the molecular alterations triggered by endothelial dysfunction largely exceeded the inherent heterogeneity of ECs. The secretomes of HCAECs and HITAECs had minor differences (65 differentially expressed proteins with a maximum logarithmic fold change of 2.33 and −2.61) despite the contrasting atherosusceptibility of these ECs. This underscores the importance of ECM production and BM assembly in endothelial physiology and indicates that suppression of transcription, translation, and protein folding exceeds both the general endothelial adaptability and differential plasticity of arterial ECs. Subsequent studies might interrogate the resilience of distinct endothelial lineages to a variety of clinically relevant endothelial dysfunction triggers. These might include high glucose (to imitate hyperglycemia, which defines diabetes mellitus and glucose intolerance), urea (to simulate the uremia observable in chronic kidney disease), lipopolysaccharide (to mimic sepsis), the S1 subunit of SARS-CoV-2 spike protein or its receptor-binding domain (to resemble the respective infection of ECs), elevated levels of pro-inflammatory cytokines (to mirror ascending grades of inflammation), mutagens (to model environmental stress), increased fatty acids such as palmitic acid (to match dyslipidaemia), or endothelial nitric oxide synthase inhibitors (to reproduce arterial hypertension scenarios).

Incubation of HCAECs and HITAECs with CPP-P or CPP-S induced the release of soluble CD59, a glycosylphosphatidylinositol-anchored glycoprotein that protects host cells from complement-mediated lysis and is ubiquitously expressed in the body, including in ECs and blood cells. In agreement with our findings, an increased expression of CD59 in ECs was observed in ECs treated with the calcineurin inhibitor cyclosporine [111], whilst CD59-positive extracellular vesicles were released in response to tumor necrosis factor alpha (TNF-α) stimulation [112]. High glucose concentrations reduced intracellular CD59 expression in concert with stimulating the release of soluble CD59 into the cell culture supernatant, suggesting CD59 shedding at glucose supersaturation [113]. Glycation of CD59 under high glucose conditions inactivated this receptor and promoted complement-driven EC lysis [114]. The potential value of soluble CD59 as an endothelial dysfunction marker requires further verification in various experimental settings and EC types.

The profiles of secreted molecules largely differed across CPP-P/CPP-S- and DPBS/MPP-treated ECs, indicating a clusterisation of physiological and pathological endothelial signatures. The former included ECM proteins (SPARC, HSPG2, CCN2, and HSP90AA1), whilst the latter contained CD59. This suggests the utility of customised 22-protein dot blotting kits for the detection of BM proteins (SPARC, HSPG2, FN1, TIMP1, LAMB1, LAMA4, LAMC1, LOXL2, NID1, FBN1, AGRN, COL4A1, COL4A2), innate immune response molecules (e.g., soluble CD59, IL-6, IL-8, MCP-1/CCL2, MIF, CXCL1, and MIP-3α/CCL20), and pro- or anti-thrombotic molecules (serpin E1/PAI-1 and uPAR) in cell culture supernatant for the detection of endothelial dysfunction in vitro. Yet, such a panel might be fine-tuned by replacing several BM molecules (e.g., those with a relatively low expression in the secretome, such as LOXL2, NID1, FBN1, AGRN, COL4A1, and COL4A2) with a number of highly expressed ECM components that do not constitute the BM (THBS1, CCN2, EFEMP1, TIMP2, MMP2, and CCN1). The dysfunctional ECs might exhibit lower release of the indicated ECM/BM molecules and higher production of pro-inflammatory cytokines and chemokines. In this experimental setting, a high abundance of the enumerated proteins would ensure reliable detection and negate the relatively low sensitivity of dot blotting, whilst the high specificity of the antibody-based applications would hold an appreciable advantage over the LC-MS approach.

The reduction in ECM proteins and increased CD59 content in the EC culture supernatant upon treatment with CPP-P or CPP-S might be explained by their pro-apoptotic effects, as a reduced number of cells produce less ECM and CD59 can enter the extracellular milieu after the degradation of the plasma membrane. In order to minimise these effects, we removed the dead cells and their remnants via centrifugation of the cell culture supernatant at 220× *g* to sediment detached cells and at 2000× *g* to sediment the cell debris before acetone-mediated protein precipitation. Yet, a certain amount of CD59 could undergo detachment from endothelial cell membranes during or after apoptosis and therefore could pass into the cell culture supernatant despite the abovementioned procedures. An elevated production of numerous pro-inflammatory cytokines and an increased CD105/endoglin content in the cell culture supernatant upon CPP-P/CPP-S treatment also supported this suggestion, although soluble CD31/PECAM1 was not augmented in these samples. Nevertheless, this should be considered a limitation of our study.

Further, we performed ELISA verification for only four differentially expressed proteins (CD59, osteonectin/SPARC, perlecan/HSPG2, and fibronectin/FN1) but not for the others, and this should be mentioned as another limitation. Yet, the proteomics approach identifies thousands of proteins in the sample in total and hundreds of differentially expressed proteins between the experimental groups, as in our study. In contrast, ELISA measures the concentration of a single protein in the cell culture supernatant. As it is virtually impossible to verify all differentially expressed proteins revealed by proteomics using ELISA, we have selected four of them that showed the most stable differential expression and are of utmost pathophysiological importance with regard to the conclusions (protectin/CD59, osteonectin/SPARC, perlecan/HSPG2, and fibronectin/FN1). As the ELISA results regarding the levels of these proteins were in accordance with the results of unbiased proteomics (UHPLC-MS/MS) analysis, we assume that these data are valid (as was evidenced by two different experimental techniques). Then, we also performed a dot blotting analysis (using an antibody-based array which permits a semi-quantitative measurement of protein levels). The relative levels of soluble endothelial markers (endoglin/CD105 and PECAM1/CD31) in the experimental groups corresponded between the dot blot and proteomics analysis, as well as between each other. Therefore, we have verified the relative expression of six proteins using two approaches, and in six out of six cases, the results of an explorative “omics” technique (UHPLC-MS/MS) were in accordance with the antibody-based verification techniques (ELISA and dot blot microarray). Collectively, we assume that these experiments provide convincing evidence that the experimental data provided in the manuscript are sufficient for valid conclusions.

Another point is that the reduced production of ECM components by the ECs after CPP treatment contradicts the molecular reprogramming observed earlier in CPP-treated cells undergoing endothelial-to-mesenchymal transition [26]. This might be partially explained by the distinct methodology, as that study focused on the transcription factors (SNAI1/Snail, SNAI2/Slug, TWIST1, ZEB1) and did not measure the ECM components [26]. Here, we did not detect the mentioned transcription factors (as they, as well as most pro-inflammatory cytokines, are secreted in relatively low amounts, which are below the sensitivity of proteomic analysis) but rather identified numerous ECM constituents that are abundantly released into the extracellular milieu by the ECs and therefore can be recognised using proteomic approaches.

The purpose of this investigation was to perform a proteome-wide profiling of the molecules secreted by the ECs into the cell culture supernatant after their treatment with primary or secondary CPPs. When comparing the 50 most abundant proteins in the pathological and physiological secretomes of HCAECs and HITAECs, we found CD59 amongst the top 50 proteins in the pathological but not the physiological secretome, regardless of the EC line. Overexpression of CD59 was stable across all samples of cell culture supernatant collected from CPP-P- or CPP-S-treated HCAECs and HITAECs. Moreover, CD59 was the only protein included in the list of the 10 most upregulated proteins in all comparisons (CPP-P vs. DPBS; CPP-P vs. MPP; CPP-S vs. DPBS; CPP-S vs. MPP) after the incubation of HCAECs and HITAECs with DPBS, MPP, CPP-P, or CPP-S (four comparisons per EC line; eight comparisons in total). Hence, soluble CD59 was abundant and significantly upregulated in the pathological secretome (i.e., in the cell culture supernatant after CPP-P or CPP-S treatment) of both HCAECs and HITAECs. We consider it important to highlight soluble CD59 as a promising biomarker of endothelial dysfunction, in particular under calcium stress. Further studies in this direction might include proteomic profiling of the secretome collected from the ECs exposed to other endothelial dysfunction triggers (e.g., high glucose, urea, lipopolysaccharide, pro-inflammatory cytokines, toxic chemicals, or fatty acids) to show whether soluble CD59 is overexpressed and basement membrane proteins are downregulated in their cell culture supernatant as well. An increase in soluble CD59 and decreased release of the ECM (in particular, basement membrane) components might reflect a general endothelial stress response rather than a CPP-specific reaction. Hence, these molecular events may not be specific to CPPs, although they certainly accompany endothelial dysfunction after the CPP treatment.

Collectively, these findings underscore the hazardous consequences of CPP internalisation, extend our understanding of endothelial response to calcium stress, and emphasize secretomic signatures of endothelial dysfunction. This might have particular importance in the context of chronic low-grade inflammation, which is characterised by elevated serum levels of inducible endothelial cytokines IL-6, IL-8, and MCP-1/CCL2 [115,116,117,118,119] and which accompanies ageing, being a typical finding in the elderly [120,121,122]. Chronic low-grade inflammation is largely determined by dysfunctional ECs [123,124,125,126], which exhibit a complex of pro-inflammatory alterations in the composition of the secretome, together defined as a senescence-associated secretory phenotype [127,128]. Such a pathophysiological pattern is frequently detected in patients with cardiovascular disease [129,130,131]. Yet, soluble CD59 has not been previously considered as either a chronic low-grade inflammation or an ageing marker [132], although CD59-mediated protection from membrane attack complex formation curbed COVID-19 progression [133].

Future studies might interrogate the role of CD59 and reduced levels of circulating BM components in the development of systemic inflammatory responses, chronic low-grade inflammation, and inflammageing. For instance, the measurement of soluble CD59 in the serum of experimental animals would be beneficial for confirming the role of this protein in endothelial dysfunction and for further analysis of its diagnostic value in this regard. The correlation of soluble CD59 with other pro-inflammatory markers (e.g., interleukin-1β or interleukin-6) in the serum also deserves a detailed evaluation in future studies. The relevant animal models that involve endothelial dysfunction include rats after the intravenous administration of CPPs or a toxic chemical, mitomycin C, rats with spontaneous arterial hypertension, hyperlipidemic mice, and aged versus young rats or mice. Patients with disturbed mineral homeostasis (e.g., hypoalbuminaemia and/or hyperphosphataemia, which are common in chronic kidney disease, chronic liver disease, or hyperparathyroidism) and/or endothelial dysfunction (e.g., elderly patients with a frailty syndrome or severe COVID-19) might be of particular interest in this context. The combination of serum proteomic profiling via LC-MS and dot blotting microarrays may verify the diagnostic and prognostic value of soluble CD59 and discover other sensitive and specific biomarkers of endothelial dysfunction.

## 4. Materials and Methods

### 4.1. Cell Culture

Primary cultures of human coronary artery endothelial cells (HCAECs, 300K-05a, Cell Applications, San Diego, CA, USA) and human internal thoracic artery endothelial cells (HITAECs, 308K-05a, Cell Applications, San Diego, CA, USA) were grown in T-75 flasks (N-708003, Wuxi NEST Biotechnology Co., Ltd., Wuxi, China) according to the manufacturer’s protocols using EndoBoost Medium (EB1, AppScience Products, Moscow, Russia) until reaching confluence. Then, HCAECs and HITAECs were subcultured into 6-well plates (N-703001, Wuxi NEST Biotechnology Co., Ltd., Wuxi, China) using 0.25% Trypsin-EDTA solution (P043p, PanEco, Moscow, Russia) and 10% foetal bovine serum (1.1.6.1, BioLot, St. Petersburg, Russia) for trypsin inhibition. Upon subculturing, HCAECs and HITAECs were grown in EndoBoost Medium (EB1, AppScience Products, Moscow, Russia) until reaching confluence (≈0.5 × 10^6^ cells per well). Immediately before the experiments, we replaced EndoBoost Medium with serum-free EndoLife Medium (EL1, AppScience Products, Moscow, Russia). During this replacement, we washed cells twice with warm (≈37 °C) Dulbecco’s phosphate-buffered saline (DPBS) without Ca^2+^ and Mg^2+^ ions (pH = 7.4, 1.2.4.7, BioLot, St. Petersburg, Russia) to remove the residual serum components that could affect further proteomic profiling or contaminate the serum-free medium with serum-derived extracellular vesicles. The rationale behind the use of these two EC lines was that coronary artery is atheroprone and the internal thoracic artery is atheroresistant. HCAECs and HITAECs were grown in parallel.

### 4.2. Artificial Synthesis and Quantification of Calciprotein Particles

To synthesise primary (CPP-P) and secondary (CPP-S) CPPs, stock solutions of CaCl_2_ (21115, Sigma-Aldrich, St. Louis, MO, USA) and Na_2_HPO_4_ (94046, Sigma-Aldrich, St. Louis, MO, USA) were diluted to equal concentrations of 3 (CPP-P) or 7.5 (CPP-S) mmol/L in Dulbecco’s modified Eagle’s medium (DMEM/F-12, 31330038, Thermo Fisher Scientific, Waltham, MA, USA) supplemented with 10% (CPP-P) or 1% (CPP-S) FBS (1.1.6.1, BioLot, St. Petersburg, Russia). For the synthesis of magnesiprotein particles (MPPs), stock solutions of MgCl_2_ (97062-848, VWR, Radnor, PA, USA) and Na_2_HPO_4_ (94046, Sigma-Aldrich, St. Louis, MO, USA) were diluted to equal concentrations of 20 mmol/L in DMEM/F-12 (31330038, Thermo Fisher Scientific, Waltham, MA, USA), supplemented with 10% FBS (1.1.6.1, BioLot, St. Petersburg, Russia). The reagents were added into DMEM/F-12 in the following order: (1) FBS; (2) CaCl_2_ or MgCl_2_; (3) Na_2_HPO_4_, with a vortexing between the added reagents. Following incubation for 24 h in cell culture conditions, the medium was centrifuged at 200,000× *g* for 1 h (Optima MAX-XP, Beckman Coulter, Brea, CA, USA), and the particle sediment was resuspended in DPBS without Ca^2+^ and Mg^2+^ ions (pH = 7.4, 1.2.4.7, BioLot, St. Petersburg, Russia).

Quantification of CPP-P, CPP-S, and MPPs was performed by a quantitation of OsteoSense 680EX-positive PKH67-negative events per µL of MPP/CPP suspension using a fluorescent probe-based flow cytometry assay. Briefly, 15 μL of the CPP suspension was added to 75 μL sterile-filtered Tris-buffered saline (pH 7.4); then, 67 μL of this mix was blended with 83 μL fluorescent-labelled bisphosphonate OsteoSense 680EX (1:75 dilution, NEV10020EX, PerkinElmer, Waltham, MA, USA) and incubated in the dark for 50 min at 4 °C, with the subsequent addition of 8.3 μL lipophilic dye PKH67 (1:100 dilution, MIDI67-1KT, Sigma-Aldrich, St. Louis, MO, USA), followed by further incubation in the dark for another 10 min at 4 °C before sample acquisition (CytoFLEX, Beckman Coulter, Brea, CA, USA). In this experimental setup, OsteoSense 680EX bound to CPPs while PKH67 discriminated CPPs from similar-sized extracellular vesicles; therefore, CPPs were defined as OsteoSense 680EX-positive PKH67-negative events. MPPs were almost devoid of calcium (≈0.03 μg/μL calcium and <10 OsteoSense 680EX-positive PKH67-negative events/μL).

The final concentration of CPP-P/CPP-S/MPP was ≈1.2 × 10^3^ particles per µL suspension. As the concentration of CPPs in the human blood was ≈2.5 × 10^5^ particles per mL, we decided to apply the dose of ≈0.6 × 10^5^ particles per mL, attributing it to ≈15–25% increase in CPP concentration, which was reported in patients with end-stage renal disease [134]. This was equal to 25 µg/µL calcium, as measured by a respective colorimetric kit (ab102505, Abcam, Cambridge, UK) at an optical density of 575 nm.

### 4.3. Sample Collection

For the secretome profiling and enzyme-linked immunosorbent assay (ELISA), confluent (≈0.5 × 10^6^ cells per well of 6-well plate) cultures of HCAECs (300K-05a, Cell Applications, San Diego, CA, USA) and HITAECs (308K-05a, Cell Applications, San Diego, CA, USA) in a serum-free EndoLife Medium (EL1, AppScience Products, Moscow, Russia) were incubated with 100 µL of MPPs, CPP-P, or CPP-S (0.6 × 10^5^ particles per mL or 25 µg/µL calcium) or an equal volume of DPBS without Ca^2+^ and Mg^2+^ ions (pH = 7.4, 1.2.4.7, BioLot, St. Petersburg, Russia) for 24 h (*n* = 3 wells per group). Then, the cell culture supernatant was withdrawn, centrifuged at 220× *g* (5804R, Eppendorf, Hamburg, Germany) to sediment detached cells, aliquoted (*n* = 3 for the secretome profiling, *n* = 6 for the dot blotting profiling, and *n* = 12 for ELISA measurements), centrifuged at 2000× *g* (MiniSpin Plus, Eppendorf, Hamburg, Germany) to sediment the cell debris, transferred into the new tubes, and frozen at −80 °C.

### 4.4. Ultra-High Performance Liquid Chromatography-Tandem Mass Spectrometry

Upon protein precipitation by acetone (300 µL serum-free cell culture medium to 1200 µL acetone, 650501, Sigma-Aldrich, St. Louis, MO, USA), the protein pellet was resuspended in 8 mol/L urea (U5128, Sigma-Aldrich, St. Louis, MO, USA) diluted in 50 mmol/L ammonium bicarbonate (09830, Sigma-Aldrich, St. Louis, MO, USA). The protein concentration was measured via a Qubit 4 fluorometer (Q33238, Thermo Fisher Scientific, Waltham, MA, USA) with a QuDye Protein Quantification Kit (25102, Lumiprobe, Cockeysville, MD, USA) according to the manufacturer’s protocol. Protein samples (15 μg) were then incubated in 5 mmol/L dithiothreitol (D0632, Sigma-Aldrich, St. Louis, MO, USA) for 1 h at 37 °C with subsequent incubation in 15 mmol/L iodoacetamide for 30 min in the dark at room temperature (I1149, Sigma-Aldrich, St. Louis, MO, USA). Next, the samples were diluted with 7 volumes of 50 mmol/L ammonium bicarbonate and incubated for 16 h at 37 °C with 200 ng of trypsin (1:50 trypsin:protein ratio; VA9000, Promega, Madison, WI, USA). The peptides were then frozen at −80 °C for 1 h and desalted with stage tips (Tips-RPS-M.T2.200.96, Affinisep, Le Houlme, France), according to the manufacturer’s protocol, using methanol (1880092500, Sigma-Aldrich, St. Louis, MO, USA), acetonitrile (1000291000, Sigma-Aldrich, St. Louis, MO, USA), and 0.1% formic acid (33015, Sigma-Aldrich, St. Louis, MO, USA). Desalted peptides were dried in a centrifugal vacuum concentrator (HyperVAC-LITE, Gyrozen Co., Ltd., Gimpo, Republic of Korea) for 3 h and finally dissolved in 20 μL 0.1% formic acid for the further shotgun proteomics analysis.

Approximately 500 ng of peptides were used for shotgun proteomics analysis using UHPLC-MS/MS with ion mobility in a TimsToF Pro mass spectrometer (Bruker Daltonics, Billerica, MA, USA) with a nanoElute UHPLC system (Bruker, Daltonics, Germany). UHPLC was performed in a one-column separation mode with a Bruker FIFTEEN separation column (C18 stationary phase, length × ID 150 mm × 0.075 mm, bead size 1.9 μm, pore size 120 Å; Bruker Daltonics, Germany) in gradient mode, with a 400 nL/min flow rate and column temperature at 50 °C. Phase A was water/0.1% formic acid (1000291000, Sigma-Aldrich, St. Louis, MO, USA) and phase B was acetonitrile/0.1% formic acid. The gradient was from 2% to 35% phase B for 25 min, then to 90% of phase B for 10 min, with a subsequent wash with 90% phase B for 10 min. The column was equilibrated with 4 column volumes before each sample. Parameters of the ion source for electrospray ionization: 1400 V of capillary voltage, 3 L/min N_2_ flow, and 180 °C source temperature. The mass spectrometry acquisition was performed in DDA PASEF mode in positive polarity, with the fragmentation of ions with at least two charges in the *m*/*z* range from 100 to 1800 and ion mobility range (1/K0) from 0.85 to 1.30 Vs/cm^2^.

Protein identification was performed in FragPipe software (version 21.1) using MSFragger (version 4.1), IonQuant (version 1.10.27), and Philosopher (version 5.1.0) in Windows 10 OS with Java v. 11.0.9.1. and AMD64 architecture. The search was performed according to the default LFQ-MBR DDA workflow with calibration and parameter optimization. The identification was performed using a human SwissProt proteome with the default FragPipe contaminant list (uploaded 10.08.2024; 20,468 proteins). The search parameters were as follows: parent mass error tolerance and fragment mass error tolerance were used according to FragPipe parameter optimization, with protein and peptide FDR < 1% and 0.1%, respectively. Cysteine carbamidomethylation (+57.02146) was set as a fixed modification. Methionine oxidation (+15.9949) and N-terminal acetylation (+42.0106) were set as variable modifications. The mass spectrometry proteomics data were deposited in the ProteomeXchange Consortium via the PRIDE partner repository [135] with the dataset identifier PXD055909. A reproducible code for the data analysis is available at https://github.com/Zoiret/Proteomic-profiling-of-HCAEC-and-HITAEC-secretome-after-exposure-to-calciprotein-particles (accessed on 17 September 2024).

### 4.5. Bioinformatic Analysis

Label-free quantification by peak area under the curve and spectral counts was used for further analysis in R (version 4.3.2; R Core Team, The R Foundation for Statistical Computing, Vienna, Austria, 2019) [136]. All proteins presented in all (3/3) biological replicates were identified, with the exception of DPBS and MPP groups in HCAECs (2/2 biological replicates). Only the proteins presented in ≥15/22 (≥70%) of samples were included in the further analysis to ensure data quality. Missed values were imputed using the k-nearest neighbours approach from the “impute” package (version 1.78.0) [137]. We performed the log-transformation using base 2 and quantile normalisation, with further analysis of differential expression using the “limma” package (version 3.60.6) [138]. Then, we carried out clusterisation of the samples via principal component analysis (PCA) in the “MixOmics” package (version 6.28.0) [139]. The “ggplot2” (version 3.5.1) [140] and “EnhancedVolcano” (version 1.22.0) [141] packages were used for visualisation. Differentially expressed proteins were defined as those with logarithmic fold change ≥ 1 and false discovery rate-corrected *p* value ≤ 0.05. We analysed the following comparisons in each of the EC lines: CPP-P vs. DPBS; CPP-P vs. MPP; CPP-S vs. DPBS; CPP-S vs. MPP; MPP vs. DPBS; and CPP-S vs. CPP-P. To address endothelial heterogeneity, we also compared HCAECs vs. HITAECs. The total number of unique and overlapping differentially expressed proteins within intergroup comparisons were visualised using a Venn diagram, employing a webtool developed by a VIB-UGent Center for Plant Systems Biology (Ghent, Belgium, https://bioinformatics.psb.ugent.be/webtools/Venn/, date accessed: 27 August 2024).

Secretome profiles of DPBS- and MPP-treated HCAECs and HITAECs were jointly screened for the proteins unique to each of these EC lines (i.e., those detected in all samples of cell culture supernatant collected from HCAECs but in none of the samples withdrawn from HITAECs). Physiological (i.e., those profiled in the cell culture supernatant from DPBS- and MPP-treated HCAECs and HITAECs) and pathological (i.e., those profiled in the cell culture supernatant from CPP-P- and CPP-S-treated HCAECs and HITAECs) secretomes were compared and the 50 most abundant proteins in each of these datasets (DPBS and MPP, HCAECs; DPBS and MPP, HITAECs; CPP-P and CPP-S, HCAECs; CPP-P and CPP-S, HITAECs) underwent pathway enrichment analysis, similar to the differentially expressed proteins.

Pathway enrichment analysis was performed using Gene Ontology [142,143] and Reactome [144,145] databases, screened using the Database for Annotation, Visualization and Integrated Discovery (DAVID, Laboratory of Human Retrovirology and Immunoinformatics, Frederick National Laboratory for Cancer Research, Frederick, MD, USA, https://david.ncifcrf.gov/tools.jsp, date accessed: 28 August 2024) [106,107]. For the filtration of bioinformatic pathways, we selected a count threshold (i.e., minimum count) of ≥5 differentially expressed proteins and applied an Expression Analysis Systematic Explorer (EASE) score, a conservative adjustment to the Fisher exact probability, which is calculated by removing one gene within the given category from the list and calculating the resulting Fisher exact probability for that category [146,147,148]. The EASE score is a measure automatically calculated by the DAVID database for pathway enrichment purposes [147,148]. We used an EASE score of 0.05 as a statistical significance threshold for maximum enrichment with pathways having a low number of proteins, although the false discovery rate-corrected *p* value was also calculated for convenience, and its threshold was defined as ≤0.05. The fifty most abundant proteins in the physiological and pathological secretomes, as well as differentially expressed proteins enriched in arterial homeostasis pathways, were plotted on a heat map.

To analyse the profile of interacting differentially expressed proteins in HCAECs and HITAECs, we applied a Search Tool for the Retrieval of Interacting Genes/Proteins (STRING) (version 12.0, https://string-db.org/?_ga=2.262501388.143005778.1725628609-1106413130.1725628609, date accessed: 26 August 2024) [149] utilising the following workflow: (1) setting the working parameters: network type (full STRING network), required score of 0.900 (highest confidence), and false discovery rate stringency of 1 percent (high); (2) filtration of differentially expressed proteins having ≥1 interaction; (3) colour mapping in order to allocate interacting differentially expressed proteins into distinct functional categories composed of closely related molecular terms; (4) pathway enrichment analysis for interacting proteins using Gene Ontology and Reactome; and (5) selection of pathways enriched with the interacting proteins and relevant for arterial homeostasis.

### 4.6. Dot Blotting Profiling

The levels of pro-inflammatory cytokines in the cell culture medium were measured by dot blotting using a Proteome Profiler Human XL Cytokine Array Kit (ARY022B, R&D Systems, Minneapolis, MN, USA) according to the manufacturer’s protocol. Chemiluminescence detection of dot blotting results was performed using an Odyssey XF imaging system (LI-COR Biosciences, Lincoln, NE, USA). Densitometric quantification was performed using ImageJ software (version 1.54k, National Institutes of Health, Bethesda, MN, USA). To increase dot blotting sensitivity, the cell culture medium was enriched using a centrifugal vacuum concentrator (HyperVAC-LITE, Gyrozen Co., Ltd., Gimpo, Republic of Korea) before the measurements. All cell culture supernatant samples were enriched equally (6-fold, from 6 mL to 1 mL).

### 4.7. Enzyme-Linked Immunosorbent Assay (ELISA)

The levels of soluble CD59, osteonectin (secreted protein acidic and rich in cysteine, SPARC), perlecan (heparan sulfate proteoglycan 2, HSPG2), and fibronectin (FN1) in the pre-centrifuged, serum-free cell culture supernatant (2000× *g*) from DPBS-, MPP-, CPP-P, or CPP-S-treated HCAECs and HITAECs were measured via ELISA (*n* = 12 per group) using the respective kits (ab263893, ab220654, ab274393, and ab219046, Abcam, Cambridge, UK) according to the manufacturer’s protocols. Colorimetric analysis was conducted using Multiskan Sky microplate spectrophotometer (Thermo Fisher Scientific, Waltham, MA, USA).

### 4.8. Statistical Analysis

Statistical analysis was performed using GraphPad Prism 8 (GraphPad Software, San Diego, CA, USA). For descriptive statistics, data are presented as median, 25th and 75th percentiles, and range. Four independent groups were compared using the Kruskal–Wallis test with Dunn’s multiple comparisons test. Adjusted *p* values ≤ 0.05 were regarded as statistically significant.

## 5. Conclusions

The treatment of HCAECs and HITAECs with CPP-P or CPP-S leads to the reduced release of ECM components and specifically BM constituents, including osteonectin (SPARC), perlecan (HSPG2), and fibronectin (FN1). In addition, incubation of the ECs with CPP-P or CPP-S results in elevated levels of soluble CD59, an 18–20 kDa membrane-bound glycoprotein that protects cells from lysis by the membrane attack complex. Taken together, these results suggest that the endothelial secretome under calcium stress is characterised by an increased CD59 shedding and an underrepresentation of ECM proteins. Further studies might investigate whether the other triggers of endothelial dysfunction also promote the detachment of CD59 and/or suppress ECM production by various EC types.

## Figures and Tables

**Figure 1 ijms-25-11382-f001:**
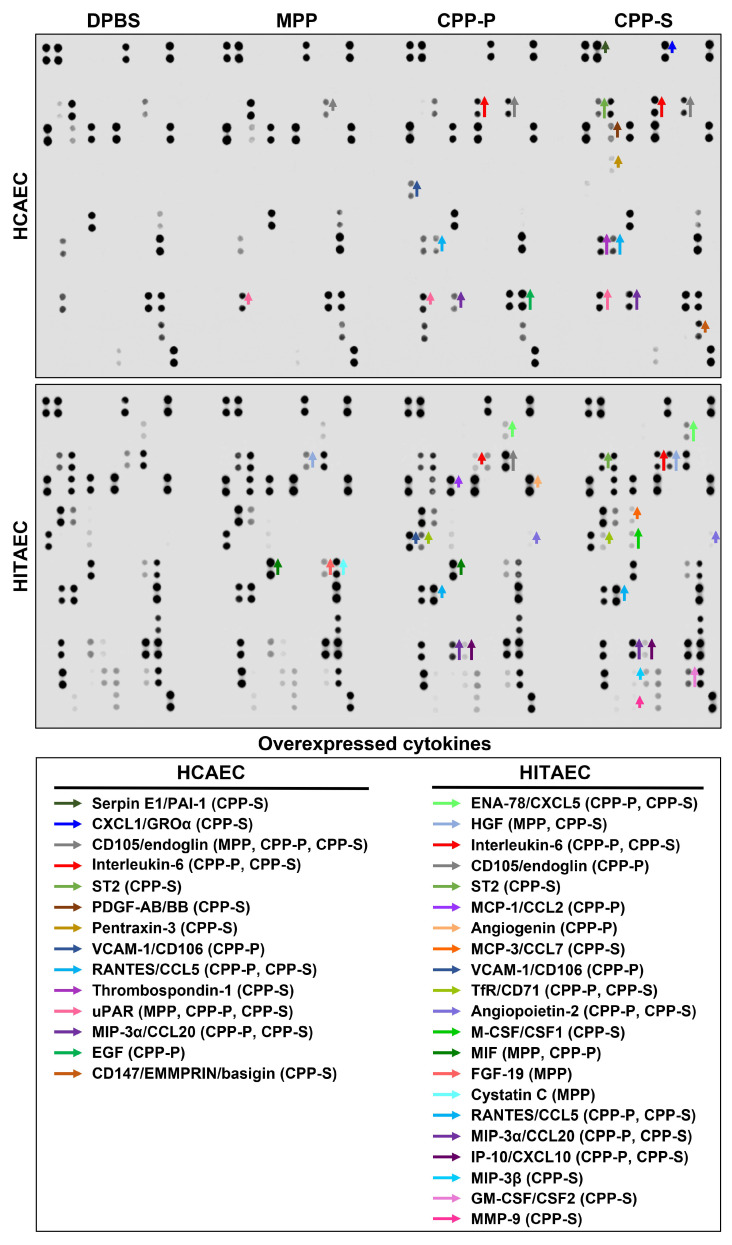
Cytokine profiling of the enriched (6-fold), serum-free cell culture medium withdrawn from HCAECs (**top**) and HITAECs (**bottom**) treated with control DPBS, magnesiprotein particles (MPP), primary calciprotein particles (CPP-P), or secondary calciprotein particles (CPP-S) for 24 h. Specific dot blotting kits for the measurement of cytokines. Below are the following colours that demarcate the signal from the respective antibodies which indicate overexpressed cytokines in the respective experimental groups: dark green: serpin E1/plasminogen activator inhibitor 1 (PAI-1); blue: chemokine (C-X-C motif) ligand 1/growth regulated protein alpha (CXCL1/GROα); gray: CD105/endoglin; red: interleukin-6 (IL-6); apple green: soluble interleukin 1 receptor-like 1/suppression of tumorigenicity 2 (ST2); brown: platelet-derived growth factor AB/BB (PDGF-AB/BB); gold: pentraxin-3; light blue: vascular cell adhesion molecule 1 (VCAM-1)/cluster of differentiation 106 (CD106); aquamarine: chemokine (C-C motif) ligand 5 (CCL5)/regulated on activation, normal T cell expressed and secreted (RANTES); violet: thrombospondin-1; dark pink: urokinase plasminogen activator surface receptor (uPAR); dark violet: macrophage inflammatory protein-3 alpha/CCL20; lime green: epidermal growth factor (EGF); light brown: CD147/extracellular matrix metalloproteinase inducer (EMMPRIN)/basigin; light green: CXCL5/epithelial neutrophil-activating protein 78 (ENA-78); lavender blue: hepatocyte growth factor (HGF); light violet: monocyte chemoattractant protein 1/chemokine (C-C motif) ligand 2 (MCP-1/CCL2); peach: angiogenin; orange: MCP-3/CCL7; salad green: transferrin receptor protein 1 (TfR1)/CD71; lavender: angiopoietin-2; bright green: macrophage colony-stimulating factor (M-CSF/CSF-1); forest green: macrophage migration inhibitory factor (MIF); pink: fibroblast growth factor 19 (FGF-19); baby blue: cystatin C; purple: interferon gamma-induced protein 10 (IP-10)/CXCL10; turquoise: MIP-3β; light pink: granulocyte-macrophage colony-stimulating factor (GM-CSF/CSF-2); deep pink: matrix metalloproteinase 9 (MMP-9). HCAEC: short, medium, and long arrows indicate fold change from 1.20 to 1.34, from 1.35 to 1.49, and ≥1.50, respectively, as compared with the DPBS group. HITAEC: short, medium, and long arrows indicate fold change from 1.25 to 1.34, from 1.35 to 1.49, and ≥1.50, respectively, as compared with the DPBS group.

**Figure 2 ijms-25-11382-f002:**
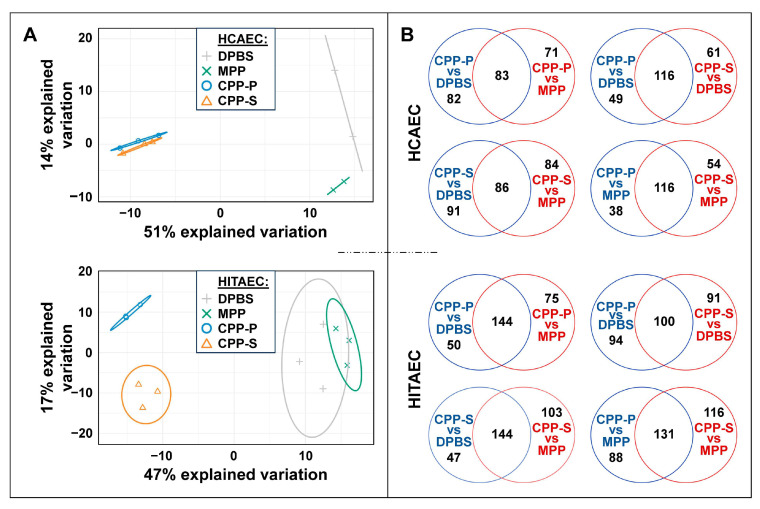
Secretome-wide comparison of protein profiles in cell culture supernatants collected from primary human coronary artery endothelial cells (HCAEC, top) and primary human internal thoracic artery endothelial cells (HITAEC, bottom) treated with Dulbecco’s phosphate-buffered saline (DPBS), magnesiprotein particles (MPP), primary calciprotein particles (CPP-P), or secondary calciprotein particles (CPP-S) for 24 h. (**A**) Principal component analysis showing the relative distance between the groups; (**B**) Venn diagrams demonstrating the numbers of differentially expressed proteins between the comparisons.

**Figure 3 ijms-25-11382-f003:**
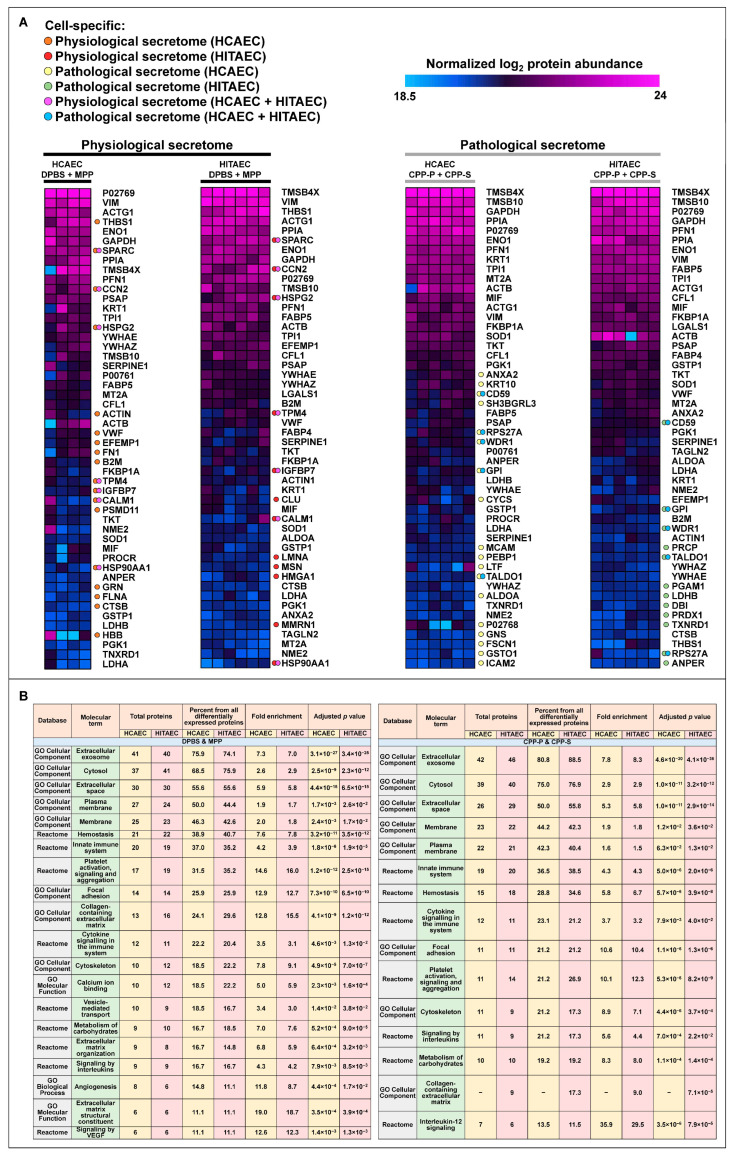
Expression and pathway enrichment analysis of protein abundance in cell culture supernatants collected from primary human coronary artery endothelial cells (HCAEC) and (**B**) primary human internal thoracic artery endothelial cells (HITAEC) treated with Dulbecco’s phosphate-buffered saline (DPBS), magnesiprotein particles (MPP), primary calciprotein particles (CPP-P), or secondary calciprotein particles (CPP-S) for 24 h. (**A**) Heat map showing the protein abundance of the 50 most abundant proteins identified in the physiological secretome (i.e., among those profiled in the cell culture supernatant from DPBS- and MPP-treated HCAECs and HITAECs, left side) and the pathological secretome (i.e., among those profiled in the cell culture supernatant from CPP-P- and CPP-S-treated HCAECs and HITAECs, right side). Protein abundance is represented as log_2_-transformed, imputed, and normalised data (see Section 4.4. in Materials and Methods “Proteomic profiling”). Orange dots highlight the proteins which are exclusively presented among the 50 most abundant in the physiological secretome in HCAECs; red dots highlight the proteins which are exclusively presented among the 50 most abundant in the physiological secretome in HITAECs; yellow dots highlight the proteins which are exclusively presented among the 50 most abundant in the pathological secretome in HCAECs; green dots highlight the proteins which are exclusively presented among the 50 most abundant in the pathological secretome in HITAECs; violet dots highlight the proteins which are exclusively presented among the 50 most abundant in the physiological secretome in both HCAECs and HITAECs; blue dots highlight the proteins which are exclusively presented among the 50 most abundant in the pathological secretome in both HCAECs and HITAECs; (**B**) Differentially expressed molecular terms were identified among these proteins during the screening of Gene Ontology and Reactome resources through the Database for Annotation, Visualization, and Integrated Discovery (DAVID).

**Figure 4 ijms-25-11382-f004:**
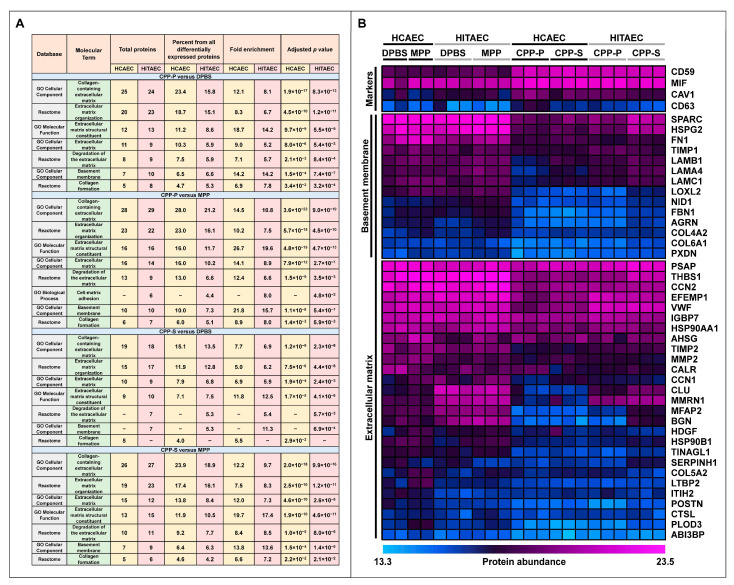
Pathway enrichment analysis of secretomes from primary human coronary artery endothelial cells (HCAEC) and primary human internal thoracic artery endothelial cells (HITAEC) treated with Dulbecco’s phosphate-buffered saline (DPBS), magnesiprotein particles (MPP), primary calciprotein particles (CPP-P), or secondary calciprotein particles (CPP-S) for 24 h. (**A**) Differentially expressed molecular terms identified among the proteins which are downregulated upon the incubation with CPP-P or CPP-S during the screening of Gene Ontology and Reactome resources through the Database for Annotation, Visualization, and Integrated Discovery (DAVID); (**B**) Heat map showing the downregulated expression of the ECM components (in particular basement membrane proteins) and notable and stable upregulation of soluble CD59 in HCAECs and HITAECs exposed to CPP-P or CPP-S. Protein abundance is represented as log_2_-transformed, imputed, and normalised data (see Section 4.4. in Materials and Methods “Proteomic profiling”).

**Figure 5 ijms-25-11382-f005:**
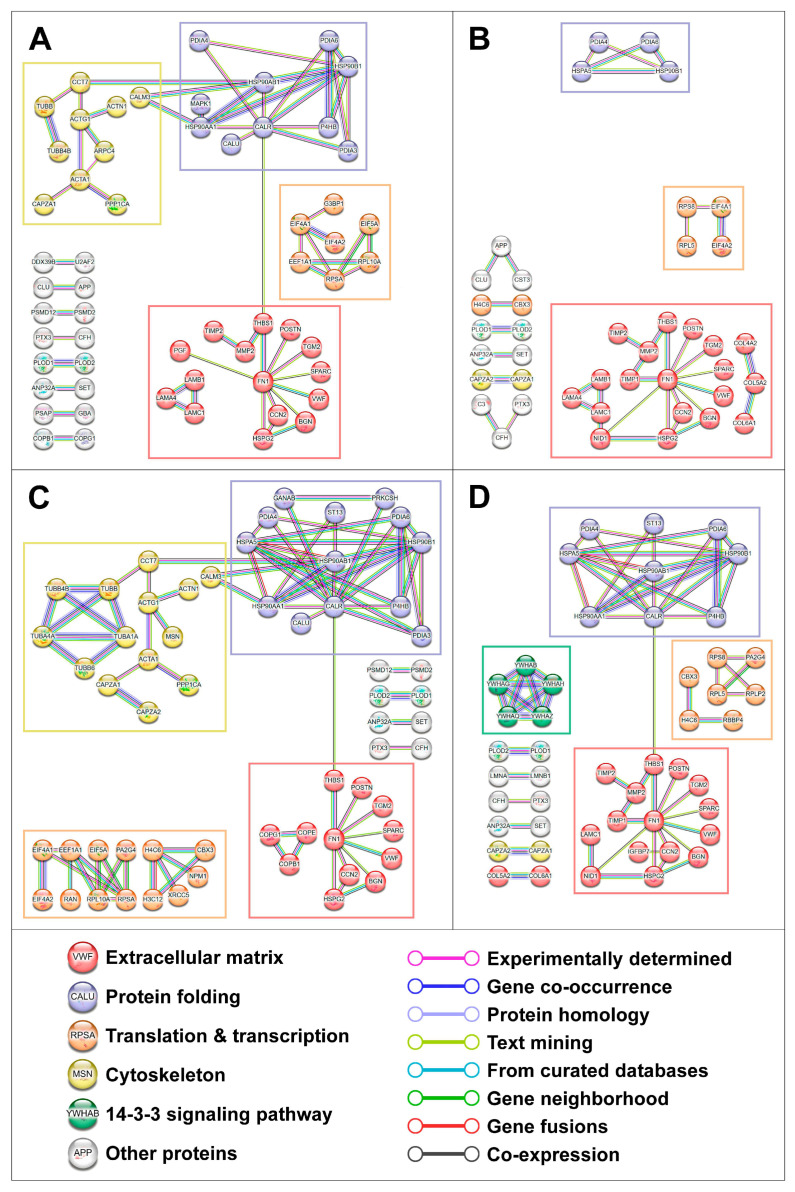
Analysis of protein–protein interactions among the downregulated proteins in secretomes from primary human coronary artery endothelial cells (HCAEC) treated with Dulbecco’s phosphate-buffered saline (DPBS), magnesiprotein particles (MPP), primary calciprotein particles (CPP-P), or secondary calciprotein particles (CPP-S) for 24 h. (**A**) CPP-P vs. DPBS; (**B**) CPP-P vs. MPP; (**C**) CPP-S vs. DPBS; (**D**) CPP-S vs. MPP. Analysis was performed using the Search Tool for the Retrieval of Interacting Genes/Proteins (STRING).

**Figure 6 ijms-25-11382-f006:**
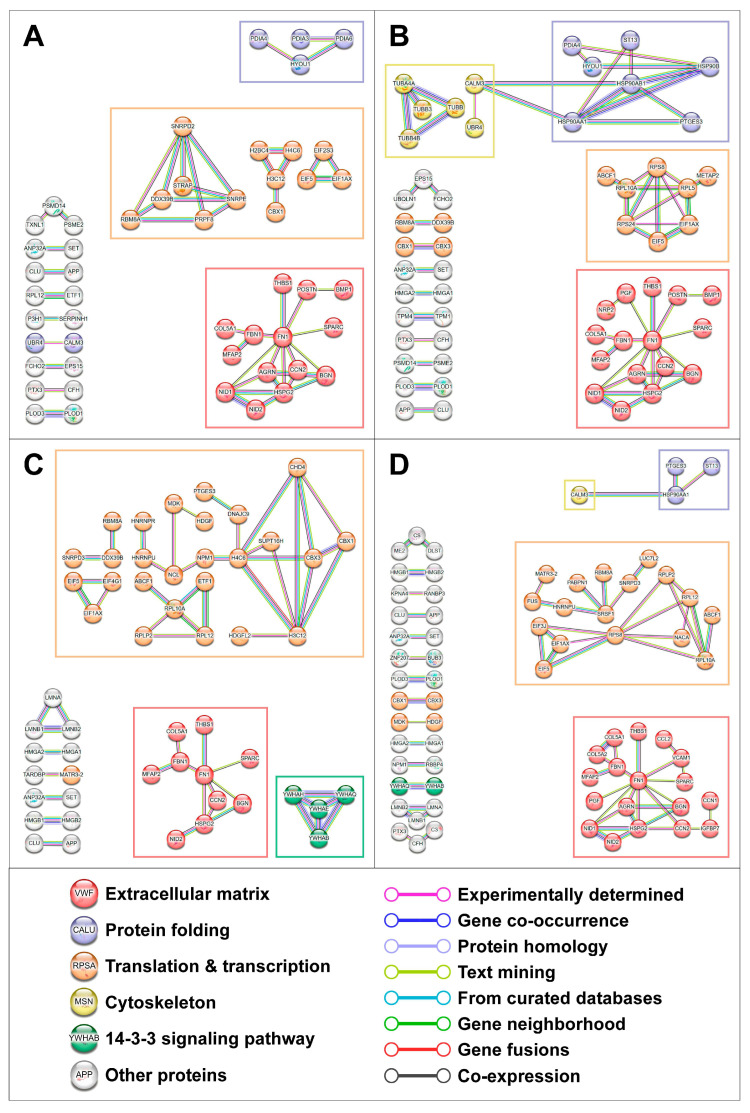
Analysis of protein–protein interactions among the downregulated proteins in secretomes from primary human internal thoracic artery endothelial cells (HITAEC) treated with Dulbecco’s phosphate-buffered saline (DPBS), magnesiprotein particles (MPP), primary calciprotein particles (CPP-P), or secondary calciprotein particles (CPP-S) for 24 h. (**A**) CPP-P vs. DPBS; (**B**) CPP-P vs. MPP; (**C**) CPP-S vs. DPBS; (**D**) CPP-S vs. MPP. Analysis was performed using the Search Tool for the Retrieval of Interacting Genes/Proteins (STRING).

**Figure 7 ijms-25-11382-f007:**
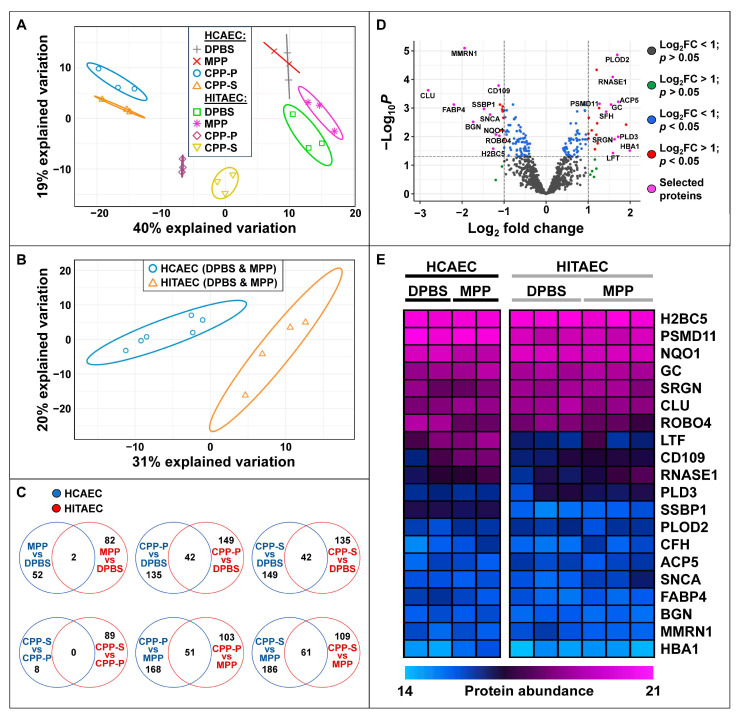
Heterogeneity of protein profiles in cell culture supernatants collected from primary human coronary artery endothelial cells (HCAEC) and primary human internal thoracic artery endothelial cells (HITAEC) treated with Dulbecco’s phosphate-buffered saline (DPBS), magnesiprotein particles (MPP), primary calciprotein particles (CPP-P), or secondary calciprotein particles (CPP-S) for 24 h. (**A**) Principal component analysis performed across all experimental groups; (**B**) Principal component analysis of pooled DPBS- and MPP-treated HCAECs and HITAECs; (**C**) Venn diagrams showing the number of overlapping and unique differentially expressed proteins between HCAECs vs. HITAECs according to the various intergroup comparisons; (**D**) Volcano plot showing upregulated and downregulated proteins between pooled DPBS- and MPP-treated HCAECs and HITAECs; the dashed lines indicate logarithmic fold change and *p* value thresholds; selected proteins are presented below on the heat map; (**E**) Heat map demonstrating protein abundance across the 10 most upregulated and the 10 most downregulated proteins between pooled DPBS- and MPP-treated HCAECs and HITAECs. Protein abundance is represented as log_2_-transformed, imputed, and normalised data (see Section 4.4. in Materials and Methods “Proteomic profiling”).

**Figure 8 ijms-25-11382-f008:**
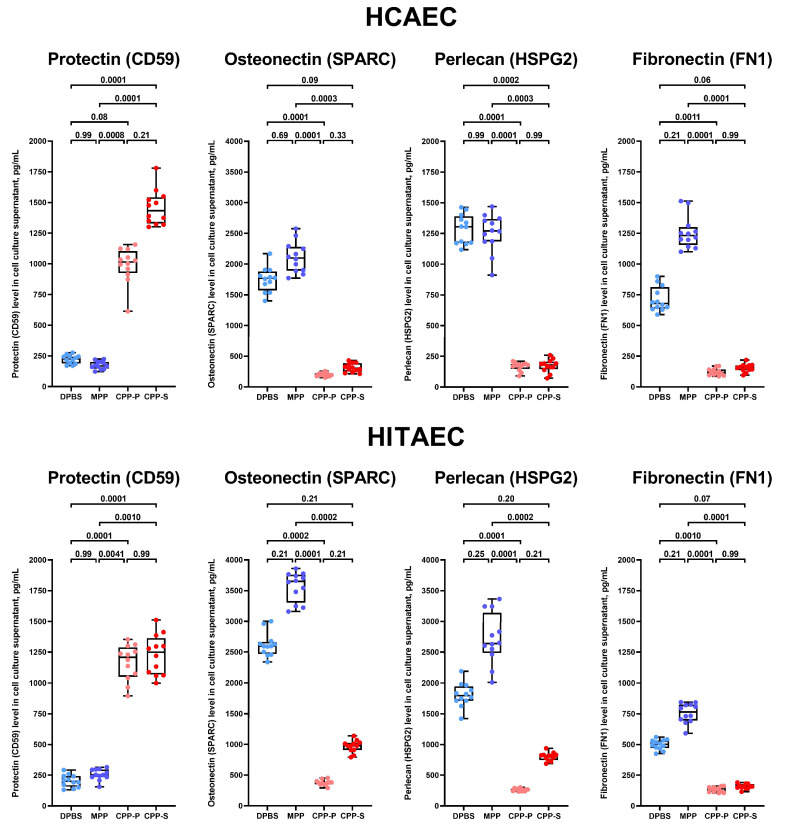
Enzyme-linked immunosorbent assay (ELISA) measurements of the levels of protectin (CD59), osteonectin (secreted protein acidic and rich in cysteine, SPARC), perlecan (heparan sulfate proteoglycan 2, HSPG2), and fibronectin (FN1) in the pre-centrifuged, serum-free cell culture supernatant (2000× *g*) from primary human coronary artery endothelial cells (HCAEC, **top**) and primary human internal thoracic artery endothelial cells (HITAEC, **bottom**) treated with Dulbecco’s phosphate-buffered saline (DPBS), magnesiprotein particles (MPP), primary calciprotein particles (CPP-P), or secondary calciprotein particles (CPP-S) for 24 h. Blue, violet, pink and red dots are for the DPBS, MPP, CPP-P, and CPP-S-treated cells, respectively. Each dot on the plots represents one measurement (*n* = 12 measurements per group). Whiskers indicate the range, box bounds indicate the 25th–75th percentiles, and centre lines indicate the median. *p* values are provided above boxes, Kruskal–Wallis test with Dunn’s multiple comparisons test.

**Table 1 ijms-25-11382-t001:** Number of overexpressed and underexpressed proteins in the secretomes of primary human coronary artery endothelial cells (HCAEC, top) and primary human internal thoracic artery endothelial cells (HITAEC, bottom) treated with Dulbecco’s phosphate-buffered saline (DPBS), magnesiprotein particles (MPP), primary calciprotein particles (CPP-P), and secondary calciprotein particles (CPP-S) for 24 h. Differentially expressed proteins were defined as those with logarithmic fold change ≥ 1 and false discovery rate-corrected *p* value ≤ 0.05. All comparisons (CPP-P vs. DPBS, CPP-P vs. MPP, CPP-S vs. DPBS, CPP-S vs. MPP, MPP vs. DPBS, and CPP-S vs. CPP-P) for each cell line were analysed.

Comparison	Number of Upregulated Proteins	Number of Downregulated Proteins
Primary human coronary artery endothelial cells (HCAEC)		
CPP-P vs. DPBS	61	104
CPP-P vs. MPP	67	87
CPP-S vs. DPBS	76	101
CPP-S vs. MPP	74	96
MPP vs. DPBS	23	31
CPP-S vs. CPP-P	3	5
Primary human internal thoracic artery endothelial cells (HITAEC)		
CPP-P vs. DPBS	73	121
CPP-P vs. MPP	87	132
CPP-S vs. DPBS	82	109
CPP-S vs. MPP	109	138
MPP vs. DPBS	17	3
CPP-S vs. CPP-P	55	34

**Table 2 ijms-25-11382-t002:** The 10 most downregulated and 10 most upregulated proteins in secretomes from primary human coronary artery endothelial cells (HCAEC) and primary human internal thoracic artery endothelial cells (HITAEC) treated with primary calciprotein particles (CPP-P) or secondary calciprotein particles (CPP-S) in comparison with those incubated with Dulbecco’s phosphate-buffered saline (DPBS) or magnesiprotein particles (MPP), for 24 h. N/A means non-inclusion in the top 10 most downregulated or upregulated proteins, but not the absence of differential expression. Differentially expressed proteins were defined as those with logarithmic fold change ≥ 1 and false discovery rate-corrected *p* value ≤ 0.05. Log_2_fold change means logarithmic fold change.

Protein	UniProt ID	HCAEC (Log_2_fold Change)				HCAEC (Log_2_fold Change)				Hits in the Top 10
		CPP-P vs. DPBS	CPP-P vs. MPP	CPP-S vs. DPBS	CPP-S vs. MPP	CPP-P vs. DPBS	CPP-P vs. MPP	CPP-S vs. DPBS	CPP-S vs. MPP	
10 most downregulated proteins for each comparison										
PTX3	P26022	−3.28	−3.24	−3.28	−3.24	−3.48	−4.35	N/A	−2.52	7/8
BGN	P21810	−2.68	−2.34	−3.05	−2.71	−4.35	−5.10	N/A	−2.41	7/8
SPARC	P09486	−3.52	−3.42	−3.36	−3.26	−2.93	−3.26	N/A	N/A	6/8
HSPG2	P98160	−3.18	−2.96	−3.08	−2.87	−2.72	−3.35	N/A	N/A	6/8
THBS1	P07996	−2.53	−3.37	−2.49	−3.33	−2.61	−3.33	N/A	N/A	6/8
DKK3	Q9UBP4	−2.83	−3.16	−2.54	−2.87	N/A	N/A	N/A	N/A	4/8
MDK	P21741	N/A	N/A	N/A	N/A	−2.92	−3.47	−2.92	−3.47	4/8
FN1	P02751	−2.24	−3.37	N/A	−3.03	N/A	N/A	N/A	N/A	3/8
ACTA1	P68133	−2.57	N/A	−2.88	N/A	N/A	N/A	N/A	N/A	2/8
SET	Q01105	−2.40	N/A	−2.68	N/A	N/A	N/A	N/A	N/A	2/8
CCN2	P29279	N/A	−2.46	N/A	N/A	N/A	N/A	N/A	−2.46	2/8
SRGN	P10124	N/A	N/A	N/A	N/A	−3.28	−4.25	N/A	N/A	2/8
CLU	P10909	N/A	N/A	N/A	N/A	−3.13	N/A	−2.00	N/A	2/8
LOXL2	Q9Y4K0	N/A	N/A	N/A	N/A	−3.04	−3.29	N/A	N/A	2/8
MFAP2	P55001	N/A	N/A	N/A	N/A	−2.98	N/A	−1.81	N/A	2/8
APOB	P04114	N/A	N/A	N/A	N/A	N/A	−3.37	N/A	−3.52	2/8
EIF1AX	P47813	N/A	N/A	N/A	N/A	N/A	N/A	−2.67	−3.15	2/8
SDF4	Q9BRK5	N/A	N/A	N/A	N/A	N/A	N/A	−2.01	−2.26	2/8
APP	P05067	N/A	N/A	N/A	N/A	N/A	N/A	−1.81	−2.67	2/8
TPP1	O14773	−2.34	N/A	N/A	N/A	N/A	N/A	N/A	N/A	1/8
FSTL1	Q12841	N/A	−2.62	N/A	N/A	N/A	N/A	N/A	N/A	1/8
CLSTN1	O94985	N/A	−2.43	N/A	N/A	N/A	N/A	N/A	N/A	1/8
CALR	P27797	N/A	N/A	−2.74	N/A	N/A	N/A	N/A	N/A	1/8
RPL10A	P62906	N/A	N/A	−2.68	N/A	N/A	N/A	N/A	N/A	1/8
LTBP2	Q14767	N/A	N/A	N/A	−2.35	N/A	N/A	N/A	N/A	1/8
EHD2	Q9NZN4	N/A	N/A	N/A	−2.29	N/A	N/A	N/A	N/A	1/8
ANP32A	P39687	N/A	N/A	N/A	−2.14	N/A	N/A	−1.93	N/A	1/8
ESM1	Q9NQ30	N/A	N/A	N/A	N/A	N/A	−2.98	N/A	N/A	1/8
HMGB2	P26583	N/A	N/A	N/A	N/A	N/A	N/A	−2.02	N/A	1/8
RBM8A	Q9Y5S9	N/A	N/A	N/A	N/A	N/A	N/A	−1.78	N/A	1/8
MATR3	P43243	N/A	N/A	N/A	N/A	N/A	N/A	−1.74	N/A	1/8
ST13	P50502	N/A	N/A	N/A	N/A	N/A	N/A	N/A	−2.33	1/8
SPOCK1	Q08629	N/A	N/A	N/A	N/A	N/A	N/A	N/A	−2.26	1/8
10 most upregulated proteins for each comparison										
CD59	P13987	2.29	2.41	2.74	2.85	2.44	2.23	2.56	2.35	8/8
MT1E	P04732	N/A	N/A	N/A	N/A	2.18	2.26	2.64	2.72	4/8
TMSB10	P63313	1.87	N/A	1.91	1.89	N/A	N/A	N/A	N/A	3/8
IGKC	P01834	N/A	2.42	N/A	2.59	N/A	2.64	N/A	N/A	3/8
IGHG1	P01857	N/A	2.31	N/A	2.14	N/A	1.83	N/A	N/A	3/8
ACTB	P60709	2.64	N/A	2.69	N/A	N/A	N/A	N/A	N/A	2/8
PSAT1	Q9Y617	1.99	N/A	2.43	N/A	N/A	N/A	N/A	N/A	2/8
ALCAM	Q13740	1.75	N/A	1.63	N/A	N/A	N/A	N/A	N/A	2/8
CAV1	Q03135	1.65	2.35	N/A	N/A	N/A	N/A	N/A	N/A	2/8
IGLL5	B9A064	N/A	2.09	N/A	N/A	N/A	1.70	N/A	N/A	2/8
SHPK	Q9UHJ6	N/A	2.07	N/A	2.01	N/A	N/A	N/A	N/A	2/8
TFRC	P02786	N/A	N/A	2.02	2.42	N/A	N/A	N/A	N/A	2/8
AFP	P02771	N/A	N/A	1.67	2.30	N/A	N/A	N/A	N/A	2/8
HNRNPH2	P55795	N/A	N/A	N/A	N/A	1.90	1.90	N/A	N/A	2/8
FLOT1	O75955	N/A	N/A	N/A	N/A	1.82	1.72	N/A	N/A	2/8
DYNLL2	Q96FJ2	N/A	N/A	N/A	N/A	1.73	1.99	N/A	N/A	2/8
FTL	P02792	N/A	N/A	N/A	N/A	N/A	N/A	2.82	2.96	2/8
STC1	P52823	N/A	N/A	N/A	N/A	N/A	N/A	2.52	2.45	2/8
REEP5	Q00765	N/A	N/A	N/A	N/A	N/A	N/A	2.33	2.35	2/8
METAP1	P53582	N/A	N/A	N/A	N/A	N/A	N/A	2.20	2.38	2/8
FTH1	P02794	N/A	N/A	N/A	N/A	N/A	N/A	2.19	2.17	2/8
BAX	Q07812	N/A	N/A	N/A	N/A	N/A	N/A	2.83	2.76	2/8
MIF	P14174	1.75	N/A	N/A	N/A	N/A	N/A	N/A	N/A	1/8
DYNC1I2	Q13409	1.69	N/A	N/A	N/A	N/A	N/A	N/A	N/A	1/8
DNASE2	O00115	1.58	N/A	N/A	N/A	N/A	N/A	N/A	N/A	1/8
CHORDC1	Q9UHD1	1.55	N/A	N/A	N/A	N/A	N/A	N/A	N/A	1/8
H1-2	P16403	N/A	2.14	N/A	N/A	N/A	N/A	N/A	N/A	1/8
KRT14	P02533	N/A	2.05	N/A	N/A	N/A	N/A	N/A	N/A	1/8
KRT5	P13647	N/A	1.95	N/A	N/A	N/A	N/A	N/A	N/A	1/8
KPRP	Q5T749	N/A	1.93	N/A	N/A	N/A	N/A	N/A	N/A	1/8
RPS28	P62857	N/A	N/A	1.89	N/A	N/A	N/A	N/A	N/A	1/8
SH3BGRL	O75368	N/A	N/A	1.71	N/A	N/A	N/A	N/A	N/A	1/8
TXNDC17	Q9BRA2	N/A	N/A	1.63	N/A	N/A	N/A	N/A	N/A	1/8
IGLL5	B9A064	N/A	N/A	N/A	2.39	N/A	N/A	N/A	N/A	1/8
DDT	P30046	N/A	N/A	N/A	1.76	N/A	1.77	N/A	2.21	1/8
MEMO1	Q9Y316	N/A	N/A	N/A	1.75	N/A	N/A	N/A	N/A	1/8
GORASP2	Q9H8Y8	N/A	N/A	N/A	N/A	2.45	N/A	N/A	N/A	1/8
CD55	P08174	N/A	N/A	N/A	N/A	1.99	N/A	N/A	N/A	1/8
P01834	P01834	N/A	N/A	N/A	N/A	1.90	N/A	N/A	N/A	1/8
CD63	P08962	N/A	N/A	N/A	N/A	1.90	N/A	N/A	N/A	1/8
LGALS9	O00182	N/A	N/A	N/A	N/A	1.68	N/A	N/A	N/A	1/8
EMD	P50402	N/A	N/A	N/A	N/A	N/A	1.79	N/A	N/A	1/8
NCSTN	Q92542	N/A	N/A	N/A	N/A	N/A	N/A	2.18	N/A	1/8
TRIAP1	O43715	N/A	N/A	N/A	N/A	N/A	N/A	1.99	N/A	1/8
S100A10	P60903	N/A	N/A	N/A	N/A	N/A	N/A	N/A	2.42	1/8

## Data Availability

The mass spectrometry proteomics data are available from ProteomeXchange Consortium via the PRIDE partner repository with the dataset identifier PXD055909. The original contributions presented in the study are included in the article/Appendix A, further inquiries can be directed to the corresponding author.

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
