# Peer review of "Proteomic Profiling of Endothelial Cell Secretomes After Exposure to Calciprotein Particles Reveals Downregulation of Basement Membrane Assembly and Increased Release of Soluble CD59"

_ijms, 2024, doi:10.3390/ijms252111382_

Round 1

Reviewer 1 Report

Comments and Suggestions for Authors

This well-executed work confirms previous findings that CPPs cause endothelial cell disruption via many mechanisms. The identification of the biomarker sCD59 and the reduction of matrix proteins are what make this manuscript innovative. Although the manuscript is interesting, there are several significant issues that it brings up that must be resolved.

1. The authors haven’t performed any well-known inflammatory cytokine analysis assay in the supernatant to confirm that there is indeed a pro-inflammatory environment.

2.  There is a possibility that CPPs are causing apoptosis in endothelial cells, and that is the cause of reduced matrix protein and release of sCD59. If there is apoptosis, then it will be difficult to say that sCD59 will be a particular biomarker for this, as this can lead to the release of many proteins.

3.  The authors have mentioned in their previous publication that CPP’s lead to apoptosis of endothelial cells. I like that publication as the authors clearly show the effect of CPPs on various cell fractions. The purpose of the current manuscript is not clear, as sCD59 has been shown in other diseases as well. Hence, sCD59 is not specific to CPPs, while it may be released because of stress and apoptosis. Reduced number of cells further leads to reduced expression of matrix proteins. CD59 overexpression is known to inhibit apoptosis.

4. Alternatively, authors should analyze the plasma of mice treated with CPPs or any similar disease model to confirm that sCD59 is indeed a biomarker, but that has to be confirmed with other known biomarkers as well.

5. The current manuscript even opposes the statement given by the authors in a previous paper that CPP-induced endothelial-to-mesenchymal transition occurs as fibronectin expression and levels in the supernatant go down.

6. Fig7; It looks like MPP is increasing the levels of fibronectin and osteonectin. Is that true, although stats do not report that?

Comments on the Quality of English Language

N/A

Author Response

We thank the reviewer for the constructive criticism and valuable notes, which collectively helped us in improving the paper. We do further provide our point-by-point response to the issues raised by the reviewer. Please see the attachment.

Reviewer 2 Report

Comments and Suggestions for Authors

In the current study the authors investigated the effect of calciprotein particles on the secretome of two endothelial cell lines.

After intensive analysis of the secrotome, the authors identify increased release of CD59 and reduced release of SPARC, HSPG2, and FN1 by proteome and ELISA technique.

Major comment:

The study is technically sound and provides a new information about four proteins and their release from stressed endothelial cells.

The main limit of this study is that the authors did not provide any data about the functional consequence of their observation. The authors can easily perform with their cell culture model whether the loss of CD59 leads to an increase of complement-dependent lysis. Similarly, the authors can check the biological significance of the observed down-regulation. In the introduction there is a lot of information given to endothelial cell biology and inflammation that turned out not to be the main effect observed here. The introduction must be improved in a way that the reader can see why the proteins later identified are important. Also in the discussion the reader expects more information (and probably also more experiments) focusing on these proteins.

Minor comment:

If you report about fold change (>1) do you report about an effect size, the mean, or the medians?

The main findings on CD59, SPARC, HSPG2, and FN1 were verified by ELISA. That means that all other reported changes have not been confirmed with an independent method and thereby are not really valid. This must be mentioned as a limitation.

Author Response

(The authors gave the same response as above.)

Round 2

Reviewer 1 Report

Comments and Suggestions for Authors

No comments.

Comments on the Quality of English Language

No comments

Author Response

We sincerely thank the reviewer for the high evaluation of our endeavours with regards to this manuscript and its revision.

Reviewer 2 Report

Comments and Suggestions for Authors

I thank the authors for the improvement of the manuscript. The remaining point is the lack of transfer to a pathological model. I do not really understand why it is not possible to perform ELISA experiments on plasma as requested. The question is simply are the data important for the conclusion (then they are required) or not. As far as I understand you agree that the manuscript without these data is preliminary and uncomplete.

Author Response

We have addressed the remaining reviewer's comments. Please see the attachment.
